

# Co-expression of HIF-1α, MDR1 and LAPTM4B in peripheral blood of solid tumors

Zaira Rehman, Ammad Fahim, Attya Bhatti, Hajra Sadia and Peter John

Atta-ur-Rahman School of Applied Biosciences (ASAB), National University of Sciences and Technology (NUST), Islamabad, Pakistan

Corresponding author
Attya Bhatti, attyabhatti@gmail.com

## ABSTRACT

The hypoxic tumor microenvironment is the major contributor of chemotherapy resistance in solid tumors. One of the key regulators of hypoxic responses within the cell is the hypoxia inducible factor-1α (HIF-1α) that is involved in transcription of genes promoting cell survival and chemotherapy resistance. Multidrug resistance gene-1 (MDR1) and Lysosome-associated protein transmembrane 4B-35 (LAPTM4B-35) are among those notable players which augment their responses to cellular hypoxia. MDR1 is the hypoxia responsive gene involved in multidrug resistance phenotype while LAPTM4B-35 is involved in chemotherapy resistance by stabilizing HIF-1α and overexpressing MDR1. Overexpression of HIF-1α, MDR1 and LAPTM4B has been associated with poor disease outcome in many cancers when studied individually at tissue level. However, accessibility of the tissues following the course of chemotherapy for ascertaining chemotherapy resistance is difficult and sometimes not clinically feasible. Therefore, indication of hypoxic biomarkers in patient's blood can significantly alter the clinical outcome. Hence there is a need to identify a blood based marker to understand the disease progression. In the current study the expression of hypoxia associated chemotherapy resistance genes were studied in the peripheral blood lymphocytes of solid tumor patients and any potential correlation with disease progression were explored. The expression of HIF-1α, MDR1 and LAPTM4B was studied in blood of 72 breast, 42 ovarian, 32 colon and 21 prostate cancer patients through real time PCR analysis using delta cycle threshold method. The statistical scrutiny was executed through Fisher's Exact test and the Spearman correlation method. There was 12–13 fold increased in expression of HIF-1α, two fold increased in MDR1 and 13–14 fold increased in LAPTM4B mRNA level in peripheral blood of breast, ovarian, prostate and colon cancer patients. In the current study there was an association of HIF-1α, MDR1 and LAPTM4B expression with advanced tumor stage, metastasis and chemotherapy treated group in breast, ovarian, prostate and colon cancer patients. The Spearman analysis also revealed a positive linear association among HIF-1α, MDR1 and LAPTM4B in all the studied cancer patients. The elevated expression of HIF-1α, MDR1 and LAPTM4B in peripheral blood of solid tumor patients can be a predictor of metastasis, disease progression and treatment response in these cancers. However, larger studies are needed to further strengthen their role as a potential biomarker for cancer prognosis.

# INTRODUCTION

Solid tumors are characteristically known to harbor a hypoxic microenvironment which serves as a major impediment to cure by conventional radiotherapy and chemotherapeutic regimens (*Brown & Giaccia, 1998*; *Teicher, 1994*). In recent years, many targeted therapies have been proposed for cancer treatment but the promiscuous nature of cancer itself makes it difficult to treat by these targeted therapies and still chemotherapy is the main treatment option to cure cancer. Moreover, tumor hypoxia works in synergism with increased drug efflux, thus eventually becoming a daunting task for clinicians to achieve complete cure (*Koh & Powis, 2012*).

Despite the fact that hypoxia is lethal to both normal as well as tumor cells, cancer cells undergo adaptive and genetic changes for their survival and even proliferation in hypoxic conditions, outpacing their normal counterparts. These changes involve transcription of many genes under the direction of transcription factor-hypoxia inducible factor-1 (HIF-1) (*Zimna & Kurpisz, 2015*). HIF-1 is a dimeric transcription factor that is comprised of the oxygen dependent HIF-1α subunit and the constitutively expressed HIF-1β subunit. The stabilization of HIF-1α is done under hypoxic conditions as well as by the action of lysosomal-associated protein transmembrane-4 beta (LAPTM4B) through an unknown mechanism (*Meng et al., 2015*). HIF-1α upregulation aids cancerous cells in surmounting the limitations of increased demand of oxygen, glucose and other nutrients by activating HIF inducible genes namely erythropoietin (*Wang & Semenza, 1993*), vascular endothelial growth factor (VEGF) (*Manalo et al., 2005*), glycolytic enzymes aldolase A, heme Oxygenase 1, enolase 1, nitric oxide synthase, lactate dehydrogenase A , phospho-fructo kinase 1, carbonic anhydrase (CA-1X) and phosphoglycerate kinase 1 (*Benita et al., 2009*; *Semenza, 2012*) and genes conferring chemotherapy resistance namely Multidrug resistance gene-1 (MDR1) (*Badowska-Kozakiewicz, Sobol & Patera, 2017*).

Multidrug resistance gene-1 is one of the oldest and well reported genes associated with chemotherapy resistance (*Nanayakkara et al., 2018*). Increased expression of MDR1 gene in tissue samples of breast, colon, ovarian and gastric cancer patients has been associated with chemotherapy resistance in these cancers (*Krishnamachary et al., 2003*; *Parissenti et al., 1999*; *Veneroni et al., 1994*; *Xia et al., 2008*; *Zhang et al., 2008*). Overexpression of MDR1 has been also attributed through overexpression of LAPTM4B) (*Li et al., 2010*). The mechanism of this regulation is explained in the subsequent paragraph.

Lysosomal-associated protein transmembrane-4 beta is an oncogene, initially identified in liver cancer. LAPTM4B associated with mammalian 4-tetratransmembrane spanning protein superfamily, and is largely located on plasma membrane and on membranous organelles such as endosomes and lysosomes (*Shao et al., 2003*). Hence, it is involved in protein trafficking and signal transduction pathways by binding to SH3

domain containing proteins (PKC, PP2A, PI3K) through its N-terminal PXXP motif (*Li et al., 2010*). Lysosome-associated protein transmembrane 4B-35 (LAPTM4B-35) is having pronounced expression in several solid tumors as HCC (*Yang et al., 2010*), lung cancer (*Tang et al., 2014*), colon cancer (*Luo et al., 2015*), breast cancer (*Xiao et al., 2013*), prostate (*Zhang et al., 2014*) and ovarian cancer (*Yin et al., 2011*). An increased expression of LAPTM4B is found to be affiliated with altered cellular transformation, metastatic cancer progression and tumorigenesis (*Yang et al., 2010*; *Zhou et al., 2010*). Overexpression of LAPTM4B is also accompanied by chemotherapy resistance in cancerous cell lines. The mechanism of this resistance is through the activation of PI3K/AKT pathway which ultimately up-regulate the MDR1 expression in cancer cells (*Li et al., 2010*).

The expression of HIF-1α, MDR1 and LAPTM4B in individual cancers has been well reported and related to cancer progression and metastasis at tissue level however in isolated studies. However, the expression of these genes in serum of cancer patients and their potential relation with clinico-pathological features are scarcely reported. Hence, these finding necessitate investigation regarding whether there will be co-expression of these genes in peripheral blood of solid tumor patients and whether they will be associated with clinico-pathological characteristics. The applicability of such an investigation may lead to novel approach for studying cancer prognosis utilizing serum rather than actual tissues.

## METHODOLOGY

### Blood sample collection

The study subjects were identified as patients who were having either biopsy proven breast, ovarian, prostate or colon cancer, aging between 25 and 70, having no other comorbidity such as diabetes, hypertension or congenital heart disease. The patients were asked for written informed consent for their potential participation in the study. Ethical approval of the study was obtained from institutional review board of Atta-ur Rahman School of Applied Biosciences NUST (IRB-15) and Shifa International Hospital, Islamabad (shifa-ref-183-2015).

Blood samples (five mL) from 72 breast, 42 ovarian, 21 prostate and 32 colon cancer patients were collected. The samples were collected in HTS sterile vacuum collection tubes and immediately put on ice to avoid RNA degradation. The clinical characteristics of patients including stage and metastasis were obtained from pathological reports (Table 1). The staging was defined using the American Joint Committee on Cancer classification system. Patients were classified into two classes on the basis of their age (less than 50 and above 50). The patients were also segregated into two groups on the basis of tumor stage; the low stage group (I–II) and the higher stage group (III–IV). For the purpose of analysis, the patients were classified into the chemotherapy naïve (no prior chemotherapy exposure) and the chemotherapy administered group (five cycles completed). Patients who were selected for the chemotherapy administered group were given standard chemotherapy treatment for breast cancer ABVD (Adriamycin (doxorubicin), Bleomycin, Vinblastine, Dacarbazine) or CEF (cyclophosphamide,

**Table 1 Clinico-pathological features of cancerous patients enrolled in the study.**

| Clinico-pathological characteristics of patients | | Breast cancer (N) (%) | Ovarian cancer (N) (%) | Prostate cancer (N) (%) | Colon cancer (N) (%) |
|---|---|---|---|---|---|
| Age | ≤50 | 46 (63.8) | 26 (61.9) | 0 (0) | 20 (62.5) |
| | >50 | 26 (36.1) | 16 (38.0) | 21 (100) | 12 (37.5) |
| Stage | I–II | 16 (22.2) | 12 (28.5) | 6 (28.5) | 13 (40.6) |
| | III–IV | 56 (77.7) | 30 (71.4) | 15 (71.4) | 19 (59.3) |
| Metastasis | Metastatic | 51 (70.8) | 28 (66.6) | 13 (61.9) | 21 (65.6) |
| | Non-metastatic | 21 (29.1) | 14 (33.3) | 8 (38.0) | 11 (34.3) |
| Treatment | Pre-treated | 57 (79.1) | 34 (80.9) | 14 (66.6) | 19 (59.3) |
| | Treatment naïve | 15 (20.8) | 8 (19.0) | 7 (33.3) | 13 (40.6) |

**Note:**
The N shows the number of patients in each group.

epirubicin, fluorouracil), ovarian cancer (Paclitaxel, carboplatin or Docetaxel, carboplatin), prostate cancer (Docetaxel prednisone or Mitoxantrone (Novantrone) and prednisone) and colon cancer (leucovorin, dexamethasone, secouracil, Oxaliplatin). For controls, the blood samples from 60 females and 30 males with age ranging between 25 and 60 years, were collected with no documented comorbidity.

## RNA extraction and cDNA synthesis

Extraction of total RNA from whole blood was conducted using TriZol reagent (Thermo Fischer Scientific, Waltham, MA, USA). All the reactions were performed on ice in order to avoid degradation. The concentration and purity of RNA was determined through NanoDrop 2000 (Thermo Fischer Scientific, Waltham, MA, USA) and the samples with ratio A260/A280 > 1.6 were used for cDNA synthesis. For cDNA synthesis 20 µL of reaction volume was prepared by adding 100ng of RNA, 1.5 mM dNTPs, 100 µM oilgodT, 200 U reverse transcriptase, 10 U RNase inhibitor and DEPC water up-to 20 µL. The reverse transcription reaction was started at 42 °C for 60 min and terminated at 70 °C for 10 min. The cDNA was then stored at −20 °C.

## Expression analysis of HIF-1α, MDR1 and LAPTM4B

The expression analysis of HIF-1α, MDR1 and LAPTM4B genes was carried out using real time PCR analysis. Primers used for expression analysis of HIF-1α forward 5′- CGCATCTTGATAAGGCCTCT-3′, Reverse 5′- TACCTTCCATGTTGCAGACT-3′, MDR1 forward 5′- AACGGAAGCCAGAACATTCC-3′, Reverse 5′- AGGCTTCCTG TGGCAAAGAG-3′, LAPTM4B forward 5′- CCTCACTGCCAGATC-3′, reverse 5′- CTATCTGTGGCATACCT-3′ and GAPDH (internal control) forward 5′- CCCCTTCATTGACCTCAACTACA-3′, reverse 5′- CGCTCCTGGAGGATGGTG AT-3′. No template/negative controls was included for all the primers in each run. The reaction mixture comprised of 200 ng of template cDNA, 0.5 µM of each forward and reverse primer, 12.5 µL of SYBER Green master mix and nuclease free water to make up 20 µL of final volume. The thermal cycler conditions were as follows; initial denaturation at 95 °C for 10 min, followed by 40 cycles of denaturation at 95 °C for 30 s,
annealing at 60 °C for 60 s and extension at 72 °C for 45 s. All the experiments were run in triplicates and their average was accounted for analysis. Data was analyzed using comparative cycle threshold method (ΔCt) and normalized according to GAPDH expression in each sample.

### Statistical analysis

Statistical inquiry was performed using Statistical software package SPSS 21 (IBM, Armonk, NY, USA). The Fisher's Exact test was used to show the relationship between different clinico-pathological variables and expression of HIF-1α, MDR1 and LAPTM4B. Spearman correlation was performed to find the association between HIF-1α, MDR1 and LAPTM4B. The significance is defined as $p < 0.05$.

## RESULTS

### Expression of HIF-1α, MDR1 and LAPTM4B in blood of cancer patients

There was a high expression of HIF-1α observed in peripheral blood of breast (2.389 ± 0.1597), ovarian (2.647 ± 0.1541), prostate (2.689 ± 0.2272) and colon (2.369 ± 0.1810) cancer patients as compared to healthy controls (0.1806 ± 0.1236). In general, there was 12–13 fold increase in expression of HIF-1α in all the studied cancer patients. There was also an elevated expression of MDR1 observed in the blood of breast (2.717 ± 0.1145), ovarian (2.498 ± 0.1191), prostate (2.391 ± 0.2224) and colon (2.634 ± 0.1782) cancer patients compared to healthy controls (1.245 ± 0.08544) (Fig. 1). Hence there was almost 2-fold increase in expression of MDR1 in cancer patients as compared to healthy controls. The high level of LAPTM4B expression was observed in the breast (2.703 ± 0.1312), ovarian (2.849 ± 0.1501), prostate (2.567 ± 0.1786) and colon (2.688 ± 0.1949) cancer patients compared to healthy controls (0.2546 ± 0.09648) (Fig. 1). In case of LAPTM4B, there was 13–14 fold increase in expression of LAPTM4B in all cancer samples compared to normal healthy controls.

### Correlation of HIF-1α, MDR1 and LAPTM4B expression with clinical features in breast cancer

The summarized statistics of HIF-1α, MDR1 and LAPTM4B expression as well as their clinico-pathological features is shown in Table 2. HIF-1α, MDR1 and LAPTM4B expression did not have any correlation with age of patients ($p > 0.05$). Instead, they showed an association with tumor stage and metastasis ($p < 0.05$). In the advanced stage group, there were >80% of patients with higher expression of all the three genes. In the lower stage group, the high expression of HIF-1α, MDR1 and LAPTM4B was found in 31%, 37% and 56% of patients, respectively. A similar scenario was observed in the metastatic and non-metastatic groups, where in the metastatic group, >80% of patients with elevated expression while in the non-metastatic group 38%, 71% and 66% of patients were having elevated expression of HIF-1α, MDR1 and LAPTM4B, respectively. It was also observed that increased expression in patients who underwent the chemotherapy treatment at the time of sample collection as compared to the treatment

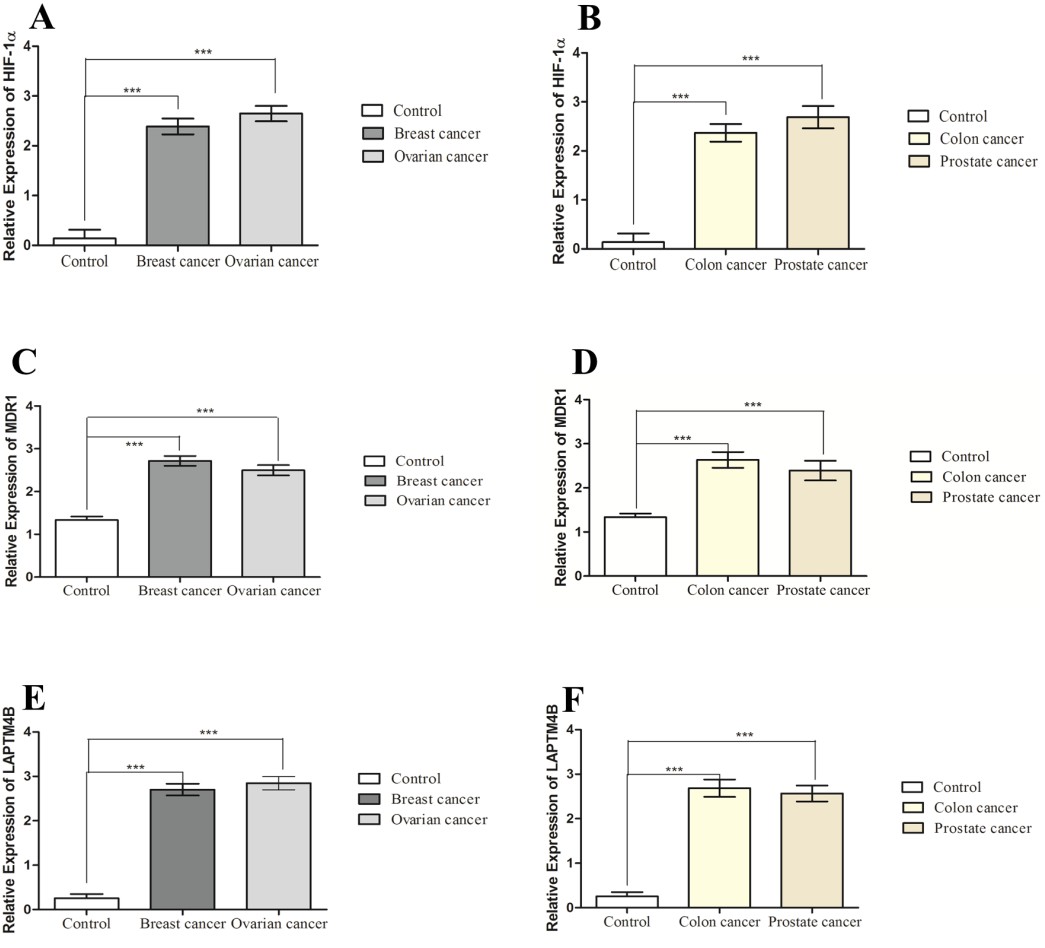

**Figure 1 Expression of HIF-1α, MDR1 and LAPTM4B.** Expression of HIF-1α in (A) breast cancer ($n = 72$) and ovarian cancer ($n = 42$) patients compared to healthy controls ($n = 50$); (B) prostate cancer ($n = 21$) and colon cancer ($n = 32$) compared to healthy controls ($n = 25$). Expression of MDR1 in (C) breast cancer ($n = 72$) and ovarian cancer ($n = 42$) patients compared to healthy controls ($n = 50$); (D) prostate cancer ($n = 21$) and colon cancer ($n = 32$) compared to healthy controls ($n = 25$). Expression of LAPTM4B in (E) breast cancer ($n = 72$) and ovarian cancer ($n = 42$) patients compared to healthy controls ($n = 50$); (F) prostate cancer ($n = 21$) and colon cancer ($n = 32$) compared to healthy controls ($n = 25$). There is increased HIF-1α, MDR1 and LAPTM4B expression in cancer patients compared to healthy controls. Representative data were presented as mean ± SEM of triplicate experiments. Statistical significance was measured by student's $t$-test (***$p < 0.001$).

naïve group (Fig. 2). 80%, 82% and 77% of patients in the pre-treated group are having elevated expression of HIF-1α, MDR1 and LAPTM4B, respectively. In treatment naïve group, 53%, 33% and 60% of patients were having high expression of HIF-1α, MDR1 and LAPTM4B, respectively.

## Correlation of HIF-1α, MDR1 and LAPTM4B expression with clinical features in ovarian cancer

The correlation of HIF-1α, MDR1 and LAPTM4B expression with clinical features of ovarian cancer patients is shown in Table 3. No correlation of HIF-1α, MDR1 and

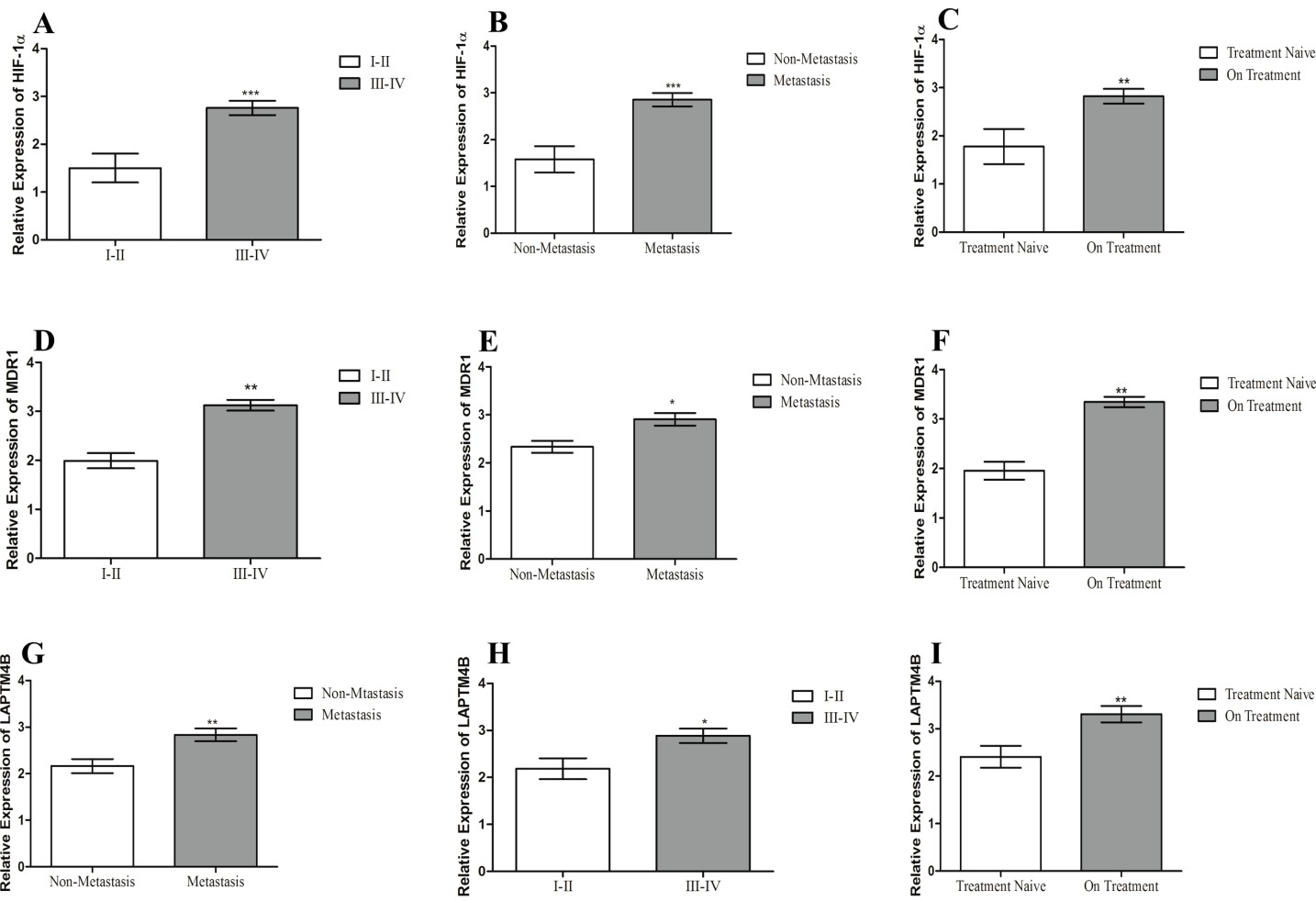

**Figure 2 Correlation of HIF-1α, MDR1 and LAPTM4B expression with clinical features of breast cancer.** Correlation of HIF-1α expression with (A) tumor stage, (B) metastasis, and (C) chemotherapy treatment in breast cancer patients. Correlation of MDR1 expression with (D) tumor stage, (E) metastasis, and (F) chemotherapy treatment in breast cancer patients. Correlation of LAPTM4B overexpression with (G) tumor stage, (H) metastasis, and (I) chemotherapy treatment in breast cancer patients. There is an elevated expression of HIF-1α, MDR1 and LAPTM4B in the metastatic, higher tumor stage and pre-treated groups compared to non-metastatic, lower tumor stage and treatment naïve groups. Representative data were presented as mean ± SEM of triplicate experiments. Statistical significance was measured by Fisher's Exact test ($^*p < 0.05$; $^{**}p \leq 0.005$; $^{***}p < 0.001$).

LAPTM4B expression was associated with age of patients in ovarian cancer ($p > 0.05$). Increased expression of HIF-1α, MDR1 and LAPTM4B was observed with higher tumor stage and distant metastasis ($p < 0.05$). Among advanced stage group, 80% of patients were presenting high expression while in the lower stage group 40–60% of patients were exhibiting high expression of three genes. In the non-metastatic group, 60% of patients with high expression of three genes while among metastatic group, 85%, 67% and 75% of patients were having elevated expression of HIF-1α, MDR1 and LAPTM4B, respectively. We observed an elevated expression of these genes in pre-treated group ($p < 0.05$) (Fig. 3). In pre-treated group 70–85% of patients, while in treatment naïve group 25-37% of patients, were having elevated expression of all three genes.

**Table 2  Correlation of HIF-1α, MDR1, and LAPTM4B expression with clinico-pathological features of Breast cancer.**

| Clinical-pathological Characteristics | | ALL N (%) | HIF-1α expression | | | MDR1 expression | | | LAPTM4B expression | | |
|---|---|---|---|---|---|---|---|---|---|---|---|
| | | | Low N (%) | High N (%) | p-value | Low N (%) | High N (%) | p-value | Low N (%) | High N (%) | p-value |
| Age | | | | | 0.66 | | | 0.066 | | | 0.140 |
| | <50 | 46 (63.8) | 16 (34.7) | 20 (43.4) | | 13 (28.2) | 33 (71.7) | | 14 (30.4) | 32 (69.5) | |
| | >50 | 26 (36.1) | 5 (19.2) | 21 (80.7) | | 5 (19.2) | 21 (80.7) | | 3 (11.5) | 23 (88.4) | |
| Stage | | | | | 0.003 | | | 0.001 | | | 0.010 |
| | I–II | 16 (22.2) | 11 (68.7) | 5 (31.2) | | 10 (62.5) | 6 (37.5) | | 7 (43.7) | 9 (56.2) | |
| | III–IV | 56 (77.7) | 10 (17.8) | 46 (82.1) | | 8 (14.2) | 48 (85.7) | | 10 (17.8) | 46 (82.1) | |
| Metastasis | | | | | 0.017 | | | 0.001 | | | 0.001 |
| | Metastatic | 51 (70.8) | 9 (17.6) | 42 (82.3) | | 12 (23.5) | 39 (76.4) | | 10 (19.6) | 41 (80.3) | |
| | Non-metastatic | 21 (29.1) | 12 (57.1) | 9 (38.0) | | 6 (28.5) | 15 (71.4) | | 7 (33.3) | 14 (66.6) | |
| Chemotherapy status | | | | | 0.007 | | | 0.021 | | | 0.001 |
| | Treatment naive | 15 (20.8) | 7 (46.6) | 8 (53.3) | | 10 (66.6) | 5 (33.3) | | 8 (53.3) | 9 (60.0) | |
| | Pre-treated | 57 (79.1) | 11 (19.2) | 46 (80.7) | | 8 (14.0) | 49 (82.4) | | 9 (15.8) | 48 (77.2) | |

Notes:
The N shows the number of patients in each group. Where p value describes the significance and p < 0.05 shows the significance of results.

**Table 3  Correlation of HIF-1α, MDR1, and LAPTM4B expression with clinico-pathological features of ovarian cancer.**

| Clinical-pathological Characteristics | | ALL N (%) | HIF-1α expression | | | MDR1 expression | | | LAPTM4B expression | | |
|---|---|---|---|---|---|---|---|---|---|---|---|
| | | | Low N (%) | High N (%) | p-value | Low N (%) | High N (%) | p-value | Low N (%) | High N (%) | p-value |
| Age | | | | | 0.700 | | | 0.181 | | | 0.115 |
| | <50 | 26 (61.9) | 7 (26.9) | 19 (73.0) | | 11 (42.3) | 15 (57.6) | | 9 (34.6) | 17 (65.3) | |
| | >50 | 16 (38.0) | 3 (18.7) | 13 (81.2) | | 4 (25.0) | 12 (75.0) | | 3 (18.7) | 13 (81.2) | |
| Stage | | | | | 0.030 | | | 0.010 | | | 0.001 |
| | I–II | 12 (28.5) | 4 (33.3) | 8 (66.6) | | 7 (58.3) | 5 (41.6) | | 6 (50.0) | 6 (50.0) | |
| | III–IV | 30 (71.4) | 6 (20.0) | 24 (80.0) | | 8 (26.6) | 22 (73.3) | | 6 (20.0) | 24 (80.0) | |
| Metastasis | | | | | 0.015 | | | 0.028 | | | 0.005 |
| | Metastatic | 28 (66.6) | 6 (21.4) | 24 (85.7) | | 9 (32.1) | 19 (67.8) | | 7 (25.0) | 21 (75.0) | |
| | Non-Metastatic | 14 (33.3) | 4 (28.5) | 8 (57.1) | | 6 (42.8) | 8 (57.1) | | 5 (35.7) | 9 (64.2) | |
| Chemotherapy Status | | | | | 0.018 | | | 0.012 | | | 0.036 |
| | Treatment naive | 8 (19.0) | 5 (62.5) | 3 (37.5) | | 5 (62.5) | 3 (37.5) | | 6 (75.0) | 2 (25.0) | |
| | pre-treated | 34 (80.9) | 5 (14.7) | 29 (85.2) | | 10 (29.4) | 24 (70.5) | | 6 (17.6) | 28 (82.4) | |

Notes:
The N shows the number of patients in each group. Where p value describes the significance and p < 0.05 shows the significance of results.

## Correlation of HIF-1α, MDR1 and LAPTM4B expression with clinical features in prostate cancer

The relationship of different clinical parameters of prostate cancer with HIF-1α, MDR1 and LAPTM4B expression is shown in Table 4 (Fig. 4). In the case of prostate cancer, all the patients were above age 50 and no patients were below 50; hence, we did not find

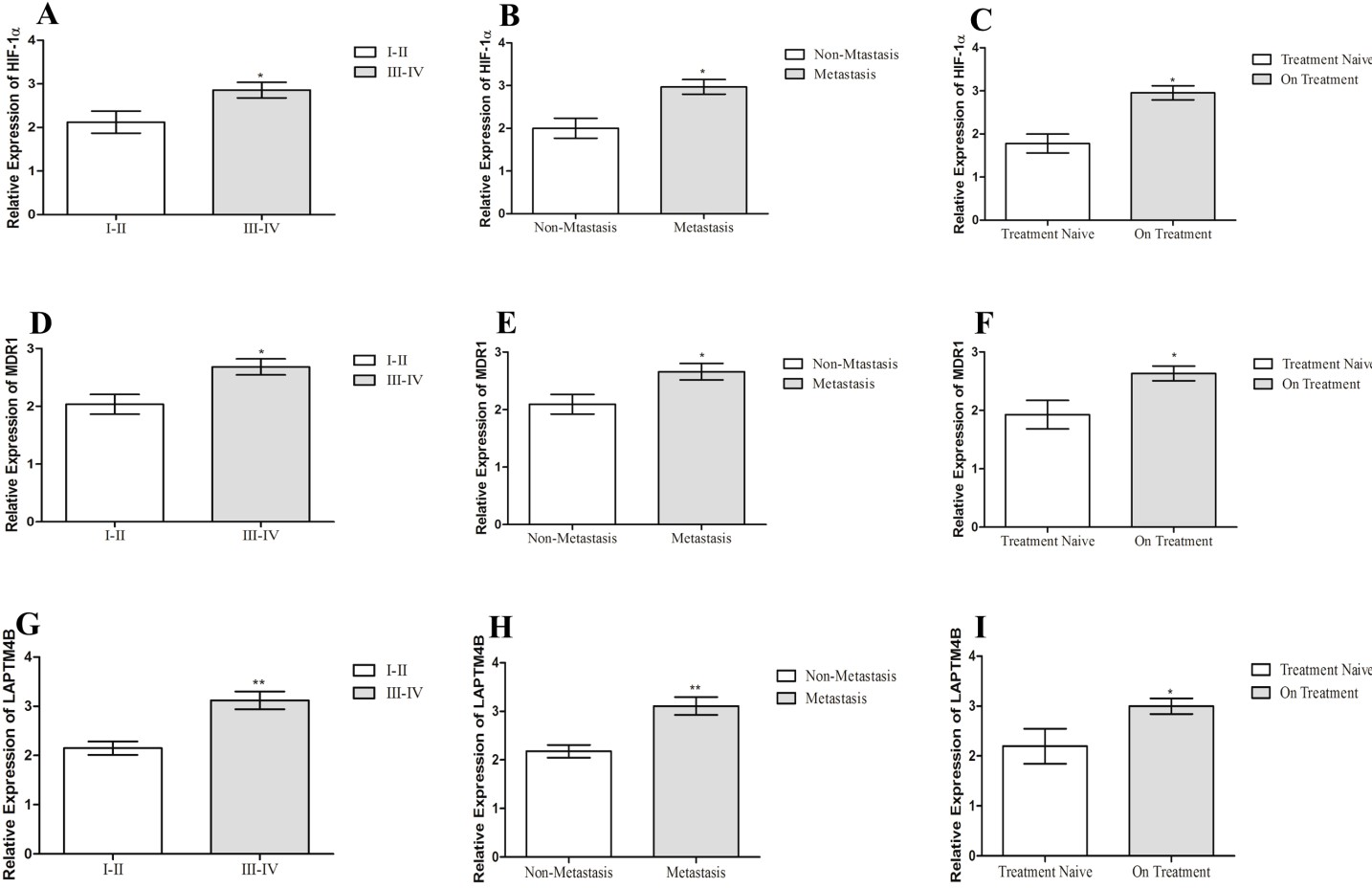

**Figure 3 Correlation of HIF-1α, MDR1 and LAPTM4B expression with clinical features of ovarian cancer.** Correlation of HIF-1α expression with (A) tumor stage, (B) metastasis, and (C) chemotherapy treatment in ovarian cancer patients. Correlation of MDR1 expression with (D) tumor stage, (E) metastasis, and (F) chemotherapy treatment in ovarian cancer patients. Correlation of LAPTM4B overexpression with (G) tumor stage, (H) metastasis, and (I) chemotherapy treatment in ovarian cancer patients. The high expression of HIF-1α, MDR1 and LAPTM4B is observed in the metastatic, high tumor stage and pre-treated groups compared to non-metastatic, lower tumor stage and treatment naïve groups. Representative data were presented as mean ± SEM of triplicate experiments. Statistical significance was measured by Fisher's Exact test (*$p \leq 0.05$).

any correlation of age with expression of HIF-1α, MDR1 and LAPTM4B. HIF-1α and LAPTM4B expression was correlated with tumor stage ($p < 0.05$). 50% of patients in lower stage group while 66% and 80% of patients in the advanced stage group exhibiting high expression of HIF-1α and LAPTM4B, respectively. Among metastatic and localized disease group, high expression of HIF-1α and LAPTM4B was observed in patients with distant metastasis ($p < 0.05$). As there were 69% and 76% of patients among metastatic group were having high expression of HIF-1α and LAPTM4B, respectively. No correlation of MDR1 expression was observed with tumor stage and metastasis ($p > 0.05$). High expression of HIF-1α and MDR1 expression was observed in the patients who were under the chemotherapy treatment ($p < 0.05$). In treatment naïve group 42% of patients while in pre-treated group, 71% and 64% of patients were having elevated expression of HIF-1α and LAPTM4B, respectively.

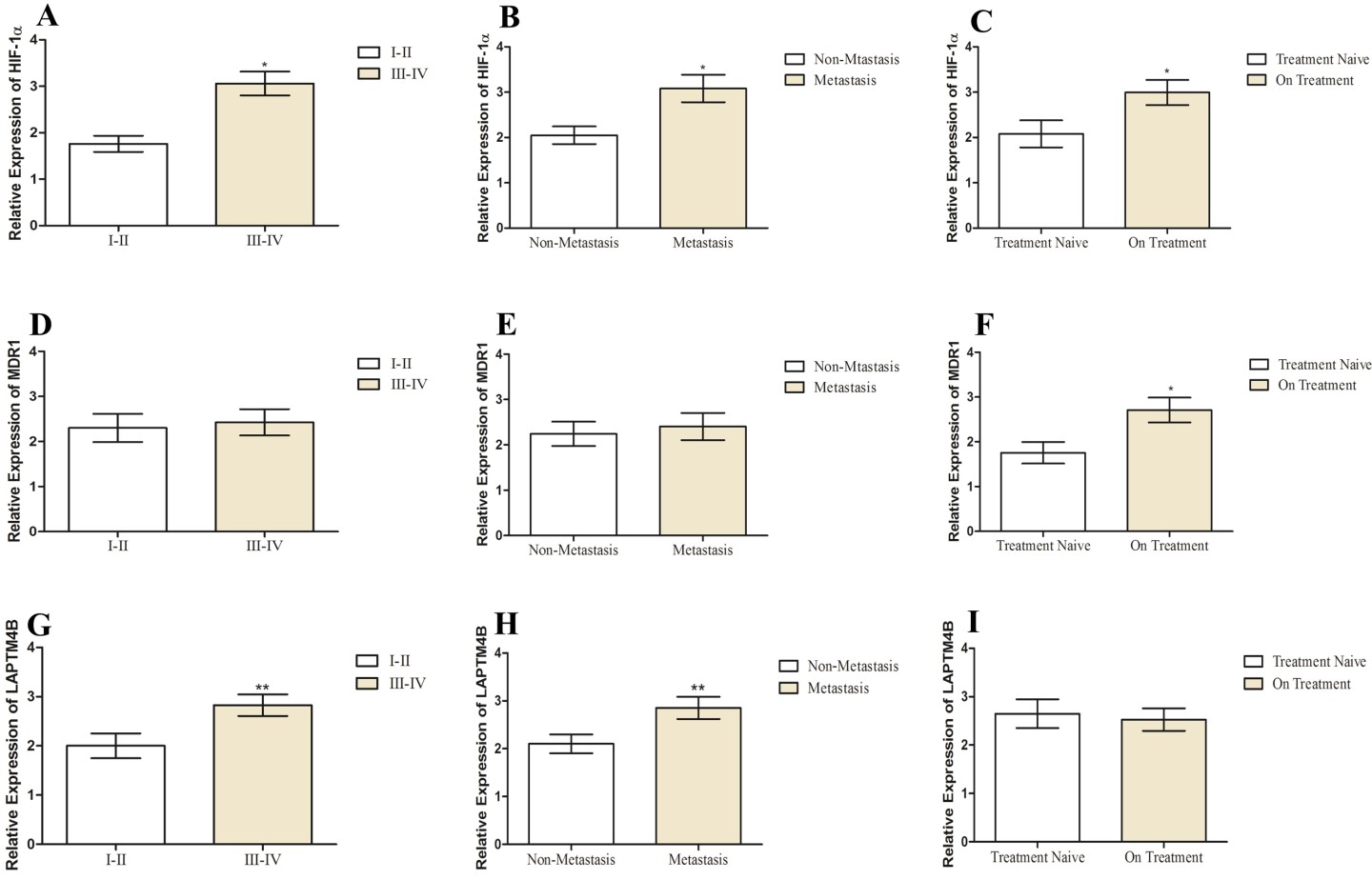

**Figure 4 Correlation of HIF-1α, MDR1 and LAPTM4B expression with clinical features of prostate cancer.** Correlation of HIF-1α expression with (A) tumor stage, (B) metastasis, and (C) chemotherapy treatment in prostate cancer patients. Correlation of MDR1 expression with (D) tumor stage, (E) metastasis, and (F) chemotherapy treatment in prostate cancer patients. Correlation of LAPTM4B overexpression with (G) tumor stage, (H) metastasis, and (I) chemotherapy treatment in prostate cancer patients. The high expression of HIF-1α and LAPTM4B is observed in the metastatic and higher tumor stage groups compared to the non-metastatic and lower tumor stage groups. No correlation is observed for MDR1 with clinical features. Representative data were presented as mean ± SEM of triplicate experiments. Statistical significance was measured by Fisher's Exact test ($^*p \leq 0.05$).

## Correlation of HIF-1α, MDR1 and LAPTM4B expression with clinical features in colon cancer

In colon cancer, high expression of HIF-1α, MDR1 and LAPTM4B was correlated to clinical features of patients (Table 5; Fig. 5). The elevated expression of HIF-1α, MDR1 and LAPTM4B was found to be correlated with advanced tumor stage and metastasis ($p < 0.05$). In the advanced tumor stage, 84% and 57% of patients, while in lower stage group 30% and 46% of patients, were having elevated expression of HIF-1α, MDR1 and LAPTM4B. There was elevated expression of all the three genes was observed in metastatic group compared to patients with non-metastatic disease as there was 85%, 62% and 52% of patients in the metastatic group were exhibiting high expression of HIF-1α, MDR1 and LAPTM4B, respectively. The elevated expression was also observed in patients who were under the chemotherapy treatment at the time of sample collection

**Table 4 Correlation of HIF-1α, MDR1, and LAPTM4B expression with clinico-pathological features of prostate cancer.**

| Clinical-pathological Characteristics | | ALL N (%) | HIF-1α expression | | | MDR1 expression | | | LAPTM4B expression | | |
|---|---|---|---|---|---|---|---|---|---|---|---|
| | | | Low N (%) | High N (%) | p-value | Low N (%) | High N (%) | p-value | Low N (%) | High N (%) | p-value |
| Stage | | | | | 0.05 | | | 0.400 | | | 0.05 |
| | I–II | 6 (28.5) | 3 (50.0) | 3 (50.0) | | 5 (83.3) | 1 (20.0) | | 3 (50.0) | 3 (50.0) | |
| | III–IV | 15 (71.4) | 5 (33.3) | 10 (66.6) | | 4 (26.6) | 11 (73.3) | | 3 (20.0) | 12 (80.0) | |
| Metastasis | | | | | 0.014 | | | 0.357 | | | 0.018 |
| | Metastatic | 13 (66.6) | 4 (30.7) | 9 (69.2) | | 6 (46.1) | 7 (53.8) | | 3 (23.0) | 10 (76.9) | |
| | Non-metastatic | 8 (33.3) | 4 (50.0) | 4 (50.0) | | 3 (37.5) | 5 (62.5) | | 3 (37.5) | 5 (62.5) | |
| Chemotherapy status | | | | | 0.029 | | | 0.048 | | | 0.476 |
| | Treatment naive | 7 (19.0) | 4 (57.1) | 3 (42.8) | | 4 (57.1) | 3 (42.8) | | 1 (14.2) | 6 (85.7) | |
| | Pre-treated | 14 (80.9) | 4 (28.5) | 10 (71.4) | | 5 (35.7) | 9 (64.2) | | 5 (35.7) | 9 (64.2) | |

Notes:
The $N$ shows the number of patients in each group. Where $p$ value describes the significance and $p < 0.05$ shows the significance of results.

($p < 0.05$). 78%, 68% and 58% of patients in the pre-treated group were showing elevated expression of HIF-1α, MDR1 and LAPTM4B, respectively.

## Association between HIF-1α, MDR1 and LAPTM4B expression in solid tumors

The Spearman analysis revealed a linear positive correlation between HIF-1α and MDR1 expression ($p < 0.001$; $R \geq 0.8$) in the blood of breast (Table 6), ovarian (Table 7), prostate (Table 8) and colon cancer (Table 9). In case of breast cancer, there were 51 patients having high expression of HIF-1α, MDR1 and LAPTM4B while 17 patients were those having low expression of all the three genes. Hence, a positive linear correlation was found between these genes with $R = 0.900$ and $p < 0.001$. In case of ovarian cancer 27 patients were those having high expression of HIF-1α, MDR1 and LAPTM4B while 10 patients were those having low expression of all the three genes, which showed a positive linear correlation among these genes ($R = 0.849$, $p < 0.001$). A similar pattern of linear correlation was also found among HIF-1α, MDR1 and LAPTM4B in case of prostate cancer ($R = 0.806$, $p < 0.001$) with 12 patients having high expression of these genes while six patients having low expression of these genes. In case of colon cancer 12 patients were those having high expression of HIF-1α and MDR1 while 15 were those with high expression of LAPTM4B also harboring high expression of MDR1 and HIF-1α. Hence a positive linear correlation was observed with $R = 0.825$ and $p < 0.001$. The positive correlation among all the studied cancers suggest similar upregulation of LAPTM4B leading to increased HIF-1α and MDR1 genes.

## DISCUSSION

Chemotherapy is one of the frontline options in administering curative or palliative care. However, drug resistant cancer phenotypes are major impediment in achieving clinical therapeutic goals. There have been practical limitations which have impacted the
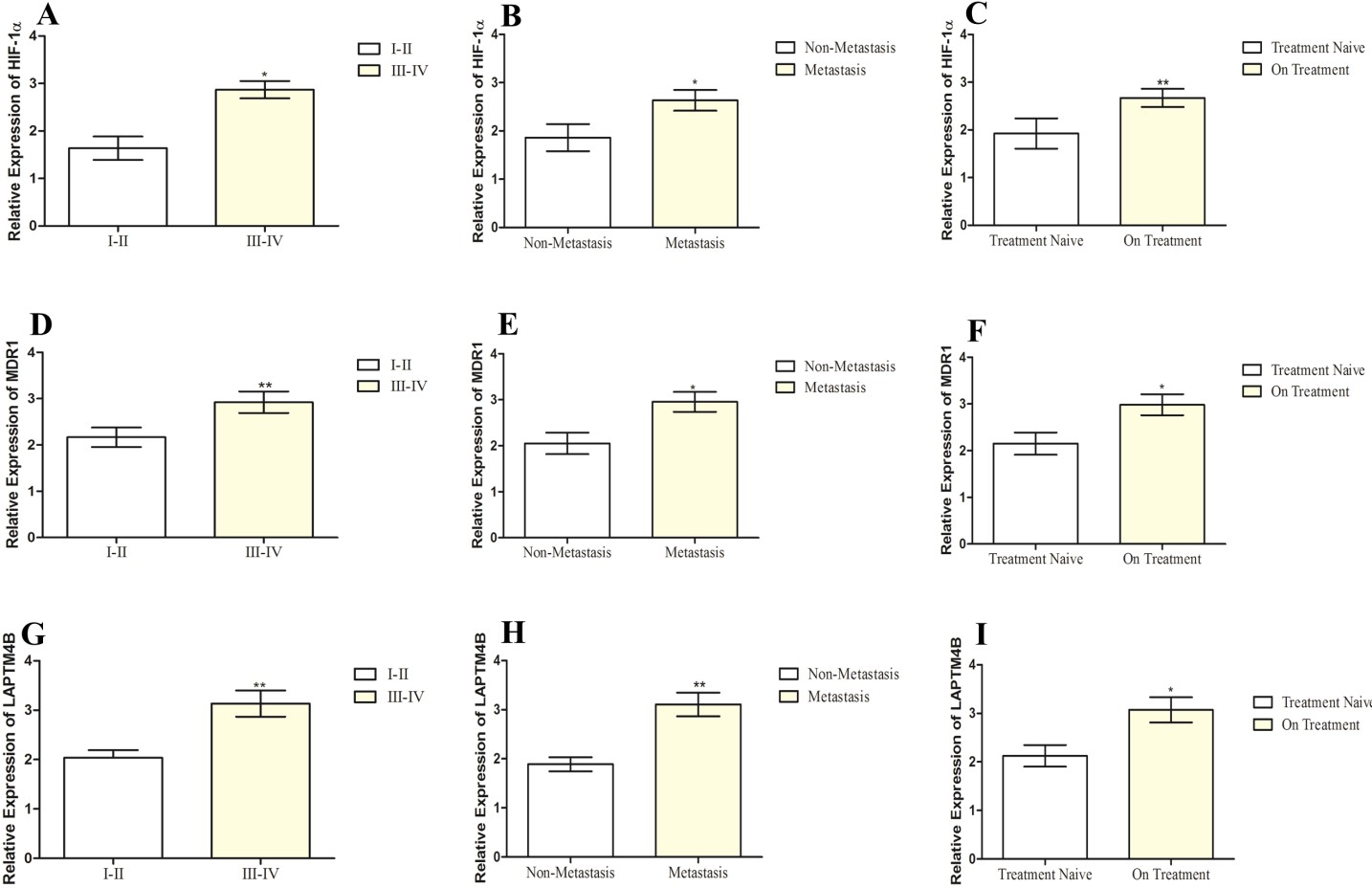

**Figure 5 Correlation of HIF-1α, MDR1 and LAPTM4B expression with clinical features of colon cancer.** Correlation of HIF-1α expression with (A) tumor stage, (B) metastasis, and (C) chemotherapy treatment in colon cancer patients. Correlation of MDR1 expression with (D) tumor stage, (E) metastasis, and (F) chemotherapy treatment in colon cancer patients. Correlation of LAPTM4B overexpression with the (G) tumor stage, (H) metastasis, and (I) chemotherapy treatment in colon cancer patients. There is elevated expression of HIF-1α, MDR1 and LAPTM4B observed in metastatic, higher tumor stage and treatment naïve group compared to the non-metastatic, lower tumor stage and pre-treated group. Representative data were presented as mean ± SEM of triplicate experiments. Statistical significance was measured by Fisher's Exact test ($^*p \le 0.05$; $^{**}p \le 0.005$).

research on identification of chemotherapy resistance markers. Generally, patients are managed by chemotherapeutic management after the combination of histological and radiological evaluation made at the time of diagnostic workup. However, as the chemotherapeutic treatment progresses, the ensuing therapy resistance processes owing to intra and inter tumoral heterogeneity may blur the prognostic outcome to negativity. Moreover, the same heterogeneity of treatment resistant sub clonal population of cancer cells, widespread metastasis and clinical complications embedded with invasive procedures decrease the feasibility of clinical reassessment by tissue re-biopsy (*Bedard et al., 2013*; *Gerlinger et al., 2012*; *Jiang et al., 2016*; *Sholl et al., 2016*).

Chemotherapy resistance has been reported in conjunction with at least three cell membrane pumps known to actively drive out chemotherapy drugs from the interior i.e MDR1, MRP1 and BCRP (*Gottesman, Fojo & Bates, 2002*; *Schinkel et al., 1994*;

**Table 5 Correlation of HIF-1α, MDR1, and LAPTM4B expression with clinico-pathological features of colon cancer.**

| Clinical-pathological Characteristics | | ALL N (%) | HIF-1α expression | | | MDR1 expression | | | LAPTM4B expression | | |
|---|---|---|---|---|---|---|---|---|---|---|---|
| | | | Low N (%) | High N (%) | p-value | Low N (%) | High N (%) | p-value | Low N (%) | High N (%) | p-value |
| Age | | | | | 0.382 | | | 0.318 | | | 0.500 |
| | <50 | 20 (62.5) | 9 (45.0) | 11 (55.0) | | 10 (50.0) | 10 (50.0) | | 11 (55.0) | 9 (45.0) | |
| | >50 | 12 (37.5) | 3 (25.0) | 9 (75.0) | | 5 (41.6) | 7 (58.3) | | 6 (50.0) | 6 (50.0) | |
| Stage | | | | | 0.016 | | | 0.005 | | | 0.007 |
| | I–II | 13 (40.6) | 7 (53.8) | 4 (30.7) | | 7 (53.8) | 6 (46.1) | | 9 (69.2) | 4 (30.8) | |
| | III–IV | 19 (59.3) | 5 (26.3) | 16 (84.2) | | 8 (42.1) | 11 (57.8) | | 8 (42.1) | 11 (57.8) | |
| Metastasis | | | | | 0.045 | | | 0.024 | | | 0.055 |
| | Metastatic | 21 (65.6) | 7 (33.3) | 18 (85.7) | | 8 (38.0) | 13 (61.9) | | 8 (38.0) | 11 (52.4) | |
| | Non-metastatic | 11 (34.3) | 5 (45.45) | 2 (18.2) | | 7 (63.6) | 4 (36.3) | | 9 (81.8) | 4 (36.4) | |
| Chemotherapy status | | | | | 0.001 | | | 0.049 | | | 0.007 |
| | Treatment naive | 13 (40.6) | 6 (46.1) | 7 (53.8) | | 9 (69.2) | 4 (30.7) | | 9 (69.2) | 4 (30.7) | |
| | Pre-treated | 19 (59.3) | 6 (31.5) | 15 (78.9) | | 6 (31.6) | 13 (68.4) | | 8 (42.1) | 11 (57.8) | |

Notes:
The N shows the number of patients in each group. Where p value describes the significance and p < 0.05 shows the significance of results.

**Table 6 Association analysis between HIF-1α, MDR1 and LAPTM4B expression in breast cancer blood specimens.**

| | | MDR1 (No. of cases) | | LAPTM4B (No. of cases) | |
|---|---|---|---|---|---|
| | | High expression | Low expression | High expression | Low expression |
| HIF-1α (No. of cases) | High expression | 51 | 0 | 51 | 0 |
| | Low expression | 3 | 18 | 4 | 17 |
| MDR1 (No. of cases) | High expression | | | 54 | 1 |
| | Low expression | | | 0 | 17 |

Notes:
The number of patients having high expression of all the three genes are highlighted in grey.
R = 0.900, p < 0.000.

**Table 7 Association analysis between HIF-1α, MDR1 and LAPTM4B expression in ovarian cancer blood specimens.**

| | | MDR1 (No. of cases) | | LAPTM4B (No. of cases) | |
|---|---|---|---|---|---|
| | | High expression | Low expression | High expression | Low expression |
| HIF-1α (No. of cases) | High expression | 27 | 5 | 30 | 2 |
| | Low expression | 0 | 10 | 0 | 10 |
| MDR1 (No. of cases) | High expression | | | 27 | 3 |
| | Low expression | | | 0 | 12 |

Notes:
The number of patients having high expression of all the three genes are highlighted in grey.
R = 0.849, p < 0.000.
**Table 8 Association analysis between HIF-1α, MDR1 and LAPTM4B expression in prostate cancer blood specimens.**

| | | MDR1 (No. of cases) | | LAPTM4B (No. of cases) | |
|---|---|---|---|---|---|
| | | High expression | Low expression | High expression | Low expression |
| HIF-1α (No. of cases) | High expression | 12 | 1 | 13 | 0 |
| | Low expression | 0 | 8 | 2 | 6 |
| MDR1 (No. of cases) | High expression | | | 12 | 3 |
| | Low expression | | | 0 | 6 |

Notes:
The number of patients having high expression of all the three genes are highlighted in grey.
$R = 0.806$, $p < 0.000$.

**Table 9 Association analysis between HIF-1α, MDR1 and LAPTM4B expression in colon cancer blood specimens.**

| | | MDR1 (No. of cases) | | LAPTM4B (No. of cases) | |
|---|---|---|---|---|---|
| | | High expression | Low expression | High expression | Low expression |
| HIF-1α (No. of cases) | High expression | 12 | 1 | 15 | 5 |
| | Low expression | 0 | 8 | 0 | 12 |
| MDR1 (No. of cases) | High expression | | | 15 | 0 |
| | Low expression | | | 2 | 15 |

Notes:
The number of patients having high expression of all the three genes are highlighted in grey.
$R = 0.825$, $p < 0.000$.

*Szakács et al., 2004*). The hyper proliferative tumor state alters biological characteristics of the tissue microenvironment inducing tumor hypoxia. This hypoxia, in turn, is one of the most important drivers of tumor aggressiveness. Normally the partial oxygen pressure ($pO_2$) ranges from between 40 and 65 mm Hg; however dropping to 10 mm of Hg or less occurs in 60% of the solid tumors (*Zimna & Kurpisz, 2015*). This hypoxia triggers HIF-1α, which is known to be a detrimental factor in further modifying tumor microenvironment towards drug resistance and aberrant tissue invasion. HIF-1α is degraded under normoxia while it stabilizes under hypoxic conditions. There are additional players contributing to HIF-1α overexpression constituting tumor suppressor genes inactivation or oncogene activation. Furthermore, v-Src, insulin, insulin-like growth factor (IGF)-1 or IGF-2, pyruvate and lactate also magnifies the HIF-1α expression (*Benita et al., 2009*; *Semenza, 2012*). The ensuing hypoxia also upregulates genes operational in lysosomal pathway which includes LAPTM4B, a novel oncogene initially identified in hepatocellular carcinoma but having important role in hypoxia induced autophagy and mitophagy (*Lai, Chang & Sun, 2016*). Previous literature suggests that LAPTM4B aids cancer cells towards survival advantage by promoting resistance to hypoxic microenvironment, nutrient deprivation and genotoxic stress contemplated by chemotherapy (*Dielschneider, Henson & Gibson, 2017*).

Previously, the role of HIF-1α, MDR1 and LAPTM4B had been individually studied in different cancers at tissue levels and provide directional evidence regarding

the engagement of these genes in tumor expansion, metastasis and therapy resistance (*Gottesman, Fojo & Bates, 2002*; *Gruber et al., 2004*; *Holzmayer et al., 1992*; *Li et al., 2010*).

The data regarding the co-expression of these genes in peripheral blood is not available. Hence, this study was aimed to measure the co-expression of HIF-1α, MDR1 and LAPTM4B genes in blood of breast, ovarian, prostate and colon cancer patients and matched with appropriate controls. To the best of our knowledge, it is the first study to put forward preliminary evidence regarding the co-expression of these genes in the blood of different solid tumors, while also finding their correlation with cancer progression.

The appearance of HIF-1α in peripheral blood may be implicated to come from PBMCs. HIF-1α expression from tumor infiltrating lymphocytes, tumor associated macrophages has been studied for its potential role in cancer progression and metastasis (*Wigerup, Påhlman & Bexell, 2016*). However, there has been reported evidence of circulating tumor cells (CTCs) to be another contributing factor of HIF-1, MDR1 and LAPTM4B in patient's blood (*Kallergi et al., 2009*; *Liu et al., 2003*; *Robey et al., 2006*). Moreover, the CTCs has also been reported to be utilized as a repeatable biomarker for monitoring tumor response (*Jakabova et al., 2017*). Serum concentrations of HIF-1α have been studies elsewhere in Graves' disease (*Liu et al., 2018*) as well as in case of prostate cancer (*Pipinikas et al., 2008*). The presence of HIF-1α can be traced back to HIF stabilization which is driven by accumulation of TCA intermediates succinate and fumarate (*Hewitson et al., 2007*; *Koivunen et al., 2007*). This has been the case along with alteration in genes encoding TCA intermediates in various solid tumors (*LaGory & Giaccia, 2016*; *Lee, Chang & Ma, 2016*; *Pollard et al., 2005*). With reference to LAPTM4B, it has been extensively studied for genotyping studies associated with various cancers from serum as well as investigated for its expression in PBMCs and platelets (*Cheng et al., 2016*; *Gnatenko et al., 2010*; *Huygens et al., 2015*; *Ma et al., 2015*; *Mo et al., 2014*).

The current study confirmed the elevated expression of HIF-1α in peripheral blood of solid tumor patients as compared to normal healthy controls that have very low or negligible expression of HIF-1α. The stabilization of HIF-1α leads to the overexpression of hypoxia responsive genes-the MDR1. The present study also confirmed the overexpression of MDR1 in the blood of solid tumor patients as compared to healthy controls. We have also found an elevated expression of LAPTM4B in blood of these patients compared to healthy controls. The results are comparable to previously published data, about the expression of HIF-1α, MDR1 and LAPTM4B studied separately in solid tumor tissue samples. The elevated HIF-1α expression has been reported in many cancers including breast, liver, colon, ovarian, brain, prostate, bladder, lung and renal cancer (*Mansour et al., 2016*; *Talks et al., 2000*; *Zhong et al., 1999*). High expression of MDR1 has been reported in ovarian (*Lu et al., 2007*), breast (*Lu et al., 2012*), non-small cell lung cancer (*Holzmayer et al., 1992*) and prostate cancer (*Bhangal et al., 2000*). Immunohistochemistry analysis also proved up-regulation of LAPTM4B-35 protein in a wide range of solid tumors (*Luo et al., 2015*; *Tang et al., 2014*; *Zhang et al., 2014*). Previously, the overexpression of LAPTM4B has been studied in serum of breast cancer patients compared to healthy controls (*Shaker et al., 2015*).
Previous findings from the reported literature as well as findings from our study direct towards potentially substantial evidence of related elevated expression of HIF-1α, MDR1 and LAPTM4B as an indicator of tumor progression, higher tumor stage and resistance to chemotherapy in breast, ovarian, colon and prostate cancer. The mechanism behind the involvement of HIF-1α in metastasis, tumor progression and chemotherapy resistance is due to the resultant overexpression of VEGF and MDR1. Both of these genes are hypoxia responsive genes (*Yang et al., 2016*). Moreover, LAPTM4B under hypoxic cellular adaptation, chip in a compelling part towards cell proliferation furthering tumor expansion rather than simply being upregulated as a secondary outcome of speedy tumorigenesis. The overexpression of LAPTM4B positively regulates autophagy and enhancing mitochondrial clearance thus preventing ROS accumulation resultantly inhibiting apoptosis (*Lai, Chang & Sun, 2016*). Moreover, the elevated expression of LAPTM4B decreases lysosomal-mediated cell death by interfering with ceramide sequestration, hence stabilizing endosomes (*Saksena et al., 2009*). Finally, LAPTM4B is known to mediate chemotherapy resistance enhancing efflux transporters (MDR1) expression (*Futter et al., 2001*).

The current study provides an evidence of close association of HIF-1α, MDR1 and LAPTM4B expression with tumor stage, metastasis and chemotherapy treatment in breast cancer patients. Previously it has been suggested that elevated HIF-1α expression in breast cancer tissue is an indicator of metastasis and relapse in breast cancer (*Bos et al., 2003*; *Generali et al., 2006*; *Gruber et al., 2004*). By doing immunohistochemistry of 740 breast cancer patients *Dales et al. (2005)*, found HIF-1α significantly correlated with metastasis and it also correlated with higher local recurrence. *Gruber et al. (2004)*, found the positive association of HIF-1α with tumor stage in breast cancer patients (*Lu et al., 2012*). Lu and colleagues identified a significant correlation of MDR1 expression with metastasis as compared to primary tumors (*Lu et al., 2012*). *Burger et al. (2003)* studied the mRNA levels in primary breast cancer patients and found that MDR1 is an important determinant of clinical outcome in patients who have been tried chemotherapy as a first line treatment option. In the current study, the patients had received the ABVD or CEF chemotherapy regimen. In the ABVD regimen doxorubicin and vinblastine were used, while in the CEF regimen epirubicin were the substrates of MDR1. Hence the patients having these regimens showed an increased level of MDR1 and LAPTM4B expression in their blood. This finding is supported by the study of *Li, Li & Zhang (1998)* that showed a high level of MDR1 and LAPTM4B expression in metastatic breast cancer correlated with resistance to anthracycline-based chemotherapy regimen. In another study, *Fan et al. (2012)* found that expression of LAPTM4B is related to risk of breast cancer development in Chinese women group.

In ovarian cancer, we observed an association of elevated HIF-1α, MDR1 and LAPTM4B expression with advance tumor stage, metastasis and pre-treated. Previously in-vitro studies in ovarian cancer cell lines, conducted by *Qin et al. (2017)*, have reported chemotherapy resistance with elevated HIF-1α expression. A meta-analysis reported by *Jin et al. (2014)* showed increased expression of HIF-1α is observed in malignant compared to benign tumors. They also reported high expression of HIF-1α in stage III–IV

along with metastasis as compared to stage I–II as well as without metastasis (*Jin et al., 2014*). A high amount of MDR1 mRNA levels in patients treated with a chemotherapeutic regimen that contains at least one P-gp substrate was reported (*Holzmayer et al., 1992*). MDR1 is involved in resistance to platinum-based chemotherapeutics in ovarian cancer cell lines (*Sonego et al., 2017*). Yin and colleagues found an association of LAPTM4B expression with stage III and chemotherapy resistance in patients who are taking PAC (cisplatin, epirubicin and cyclophosphamide) regimen in ovarian cancer (*Yin et al., 2011*). In the present study patients are receiving the platinum-based chemotherapy (cisplatin or carboplatin) hence they develop resistance to these agents. *Yang et al. (2008)* also reported that elevated expression of LAPTM4B is associated with advanced tumor stage, and it is neither correlated with age not with gender in 85 ovarian cancer tissue samples which shows that LAPTM4B overexpression is linked with ovarian cancer progression.

In prostate cancer, we observed an association of increased HIF-1α and LAPTM4B expression with advanced tumor stage and metastasis. No correlation of MDR1 was observed with tumor stage and metastasis. In case of LAPTM4B, no correlation was observed with chemotherapy treatment and this may be due to small sample size of the study. We observed increased expression of HIF-1α and MDR1 in patients who were under the chemotherapy treatment. Our results showed similar findings compared to previous studies on the expression of HIF-1α in prostate cancer. *Wang et al. (2006)* found a significant correlation of HIF-1α expression with metastasis. Fojo et al., identified the low level of P-gp expression in normal prostate tissue while *van der Valk et al. (1990)*, reported a high level (MDR1) in prostate cancer. Contrary to our results *Izbicka et al. (1998)*, observed high MDR1 mRNA expression in metastatic prostate cancer patients as compared to localized disease stage. Previously, the association of high LAPTM4B expression was observed with poor prognosis, high TNM stage, lymph node metastasis and seminal vesicle invasion in prostate cancer patients. In general high LAPTM4B expression is shown to be associated with poor prognosis in prostate cancer (*Zhang et al., 2014*).

In colonic cancer patients, a similar trend was observed where high HIF-1α, MDR1 and LAPTM4B expression was correlated with advanced tumor stage, metastasis as well as chemotherapy treatment in isolated studies. *Mansour et al. (2016)* studied the similar trend where high HIF-1α expression directs towards advance stages. *Kang et al. (2012)* studied the expression of LAPTM4B in colon cancer tissue samples and found a high expression compared to controls.

This study also analyzed any potential mutual correlation patterns observed among HIF-1α, MDR1 and LAPTM4B. The Spearman correlation analysis suggested the HIF-1α expression to be significantly associated with increased MDR1 and LAPTM4B expression in breast, ovarian, colon and prostate cancer patients. These results are consistent with previous findings where high expression of MDR1 and HIF-1α associated with tumor stage and metastasis in colon cancer and lung small cell carcinoma. Hence, these two proteins may be considered as potential candidate biomarkers for predicting the malignant progression and metastasis of human colon and LSCC (*Ding et al., 2010*; *Xie et al., 2013*). It has been reported previously that LAPTM4B regulates expression of MDR1

through activation of PI3K/AKT pathway (*Li et al., 2010*). As the synthesis of HIF-1α is also regulated through PI3K/AKT pathway hence LAPTM4B also regulated expression of HIF-1α.

Our study results were limited by the unavailability of verified survival data and lost to follow up status of the patients recruited in this study, which may have broadened the validit of our results. The ever evolving face of intra-tumoral heterogeneity has continued the quest for circulating biomarkers for cancer prognosis, however recent reports have supplement to potential consideration list the DNA biomarkers like BRAF, PIK3CA, MGMT, KRAS, TP53, circulating tumor DNA and RNA biomarkers like miR-155, miR21-5p, miR125b-5p, miR200, miR210 and miR221 (*Rapisuwon, Vietsch & Wellstein, 2016*). However, none of them have proven to be clinically conclusive for all cancers or even for all solid tumors. This study provides preliminary evidence directing towards the utilization of co-expressional profiles of HIF-1α, MDR1 and LAPTM4B as potential candidate for serum based biomarkers for tumor progression however warranting further clinical surveillance. Moreover, further studies with bigger patient cohort will be required to elaborate the prognostic role of HIF1, MDR1 and LAPTM4B in all solid tumors. Additionally, the new evidence stemming from this study contributes to the addition of our understanding of LAPTM4B in cancer development and progression. Furthermore, LAPTM4B as a mediator of hypoxia induced autophagy provides cyto-protection and contribution in evading chemotherapeutic drug response (*Li et al., 2011*), which can be further investigated as a therapeutic target in this context.

## CONCLUSION

In the current study, we identified a high level of expression of HIF-1α, MDR1 and LAPTM4B in peripheral blood of representative solid tumor patient samples. The high expression of these genes has been found to be associated with disease progression. The reflection of peripheral blood expression of these genes in solid tumor patients may serve to further establish these genes as potential circulating biomarker for cancer prognosis and therefore by attributing the functional significance of HIF-1α, MDR1 and LAPTM4B, addition of angiogenic inhibitors coupled with autophagy inhibitors may synergistically aid in the effective therapy and prevent drug resistance.

### Funding

This work was supported by Higher Education Commission 5000 Indigenous Scholarship Scheme. The funders had no role in study design, data collection and analysis, decision to publish, or preparation of the manuscript.

### Grant Disclosure

The following grant information was disclosed by the authors:
Higher Education Commission 5000 Indigenous Scholarship Scheme.

## Competing Interests

The authors declare that they have no competing interests.

## Author Contributions

- Zaira Rehman conceived and designed the experiments, performed the experiments, analyzed the data, prepared figures and/or tables, authored or reviewed drafts of the paper, approved the final draft.
- Ammad Fahim authored or reviewed drafts of the paper, approved the final draft.
- Attya Bhatti conceived and designed the experiments, contributed reagents/materials/analysis tools, authored or reviewed drafts of the paper, approved the final draft.
- Hajra Sadia conceived and designed the experiments, analyzed the data, contributed reagents/materials/analysis tools, authored or reviewed drafts of the paper, approved the final draft.
- Peter John approved the final draft.

## Human Ethics

The following information was supplied relating to ethical approvals (i.e., approving body and any reference numbers):

Shifa International Hospital (Ethical Application Ref: **shifa-ref-183-2015 IRB-15**).

Atta ur Rahman School of Applied Biosciences, National University of Sciences and Technology, Islamabad, Pakistan.

## Data Availability

Raw data are available as a Supplemental File.

## Supplemental Information

Supplemental information for this article can be found online at http://dx.doi.org/10.7717/peerj.6309#supplemental-information.

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
