# Peer review of "Co-expression of HIF-1α, MDR1 and LAPTM4B in peripheral blood of solid tumors"

_PeerJ, doi:10.7717/peerj.6309_

## Round 0.1 · original submission · Major Revisions

Although the subject is interesting and there are a number of positive remarks by the Referees, the paper cannot be accepted in its present form.

Reviewers 2 and 3 raise several criticisms and ask for major revisions, including new experiments. If the authors decide to re-submit the manuscript, along with the requested revisions, a point-by-point rebuttal letter should be provided.

·

Basic reporting

Rehman Z. et al. presented a work “Co-expression of HIF-1α, MDR1 and LAPTM4B in Peripheral Blood of Solid Tumors”. The results are well explained and the methods are really clear and reproducible. The general structure of the manuscript and the format make it readable also from experts of other medicine fields.

Experimental design

The topic fits with the scope of the journal, by focusing on biomarkers of tumor hypoxia in chemo-resistance in various cancer types, thus addressing a general question in cancer therapy/medicine, and, moreover, by using an innovative method for biomarkers on liquid biopsy, that is generally applicable. Generally, the experimental design is very simple, but well statistically conducted to be meaningful.

Validity of the findings

Authors demonstrated a significant association between HIF-1α, MDR1 and LAPTM4B and their correlation with negative prognostic factors. Results are obtained by comparison to healthy controls and are validated by statistical analysis thus reinforcing the findings.

Additional comments

Minor comments:
- The hypothesis and the gap of knowledge are not sufficiently clear in the abstract. Authors can add in line 23-26 a sentence about the advantage to find a biomarker detectable on liquid biopsy/blood sample in cancer patients (thus overcoming the issue of tissue re-biopsy) and then state their hypothesis that hypoxia biomarkers can be detected also on blood samples and introduce the correlations’ experiments.
- In introduction section, lines 46-48 should be re-paraphrased. The main concept is that targeted therapies are helping improving cancer patients’ outcome but chemotherapy still represent a very important part of anti-cancer therapy for solid tumors, including the ones analyzed in the work.
- In lines 67-69 clarify that they are results on tissue samples.
- In line 70 consider to specify “as below clarified” since this data is better explained in the subsequent text paragraph.
- In lines 88-90 authors should state clearly the hypothesis of the study based on the gap of knowledge in terms of data on blood samples for the hypoxia related biomarkers expression in solid cancers.
- In line 107, authors state that chemotherapy treated samples are “five cycles” of chemotherapy. Please clarify if it means patients underwent blood sampling after 5 cycles of chemotherapy in any cancer types independently from the chemotherapy scheme and which is the rationale for the decision of this time point (median time of clinical or radiological re-staging?).
- In line 151, there is a lacking of one+ in first parenthesis.
- In lines 192-194 please describe in details any associations found in co-expression of the three genes.
- Lines 227-229 re-phrase to ‘HIF-1α expression from TILs (Tumor Infiltrating Lymphocytes), TAMs (Tumor Associated Macrophages) has been studied for its potential role in cancer progression and metastasis (Wigerup et al. 2016).”
- Lines 227-229 re-phrase to “Finally, LAPTM4B is known to mediate chemotherapy resistance by enhancing efflux transporters (MDR1) expression”(Futter et al. 2001).
- In the tables, please consider to use “pre-treated” instead of “treatment experienced”, as this is a more diffused expression. The same expression could be used in the text.
- Add in discussion a reference on issue of re-biopsy of patients after chemotherapy (performace status? necrosis of tissue?) and the advantage of blood testing (overcome the heterogeneity inter-metastasis).
- Table 2, 3, 4, 5 should be reformatted to fit the journal printout, they are not completely visible in the merged PDF uploaded.
- Consider to use a gray color scale/re-shaping for the tables 6-9 to make more impactful the expression of any single gene and their correlations.
- Generally, some typos errors are present and should be addressed by careful correction.

Reviewer 2 ·

Basic reporting

The manuscript “Co-expression of HIF-1α, MDR1 and LAPTM4B in Peripheral Blood of Solid Tumors” by Rehman Z et al shows an elevated expression of HIF-1α, MDR1 and LAPTM4B in blood of patients with different types of solid tumors and a correlation of such expression with tumor stage, metastasis and chemotherapy treatment. With the present study, the authors want give force to the concept that the high level of the selected proteins in the blood of cancer patients could be a predictor of disease progression and treatment response.
The present study addresses a clinical research topic within scope of the journal and the proposed research question is well defined but there are some comments:
-To better support the conclusion of the study, the authors should indicate the levels of HIF-1α, MDR1 and LAPTM4B in the tumor biopsy of patients thus comparing the results with the expression of the same proteins in the peripheral blood. Furthermore, it is not clear what is the blood fraction isolated to detect the expression of the selected genes. Can the presence of circulating tumor cells give a contribution to detection of HIF-1α, MDR1 and LAPTM4B in the blood of the selected patients?

Experimental design

-Methods should be improved especially in the section referred to the collection and analysis of blood samples. It is not clear what blood fractions were isolated.
-The figures and tables should be described with more deatils both in the legends and in the text.
-Table 4 lacks the age correlation
-The tables indicating the linear correlation between several parameters should be accompanied by a graphic representation thus improving the data understanding and points distribution.
- In the figure 1C and 1D there is a difference between raw data and the value reported in the histograms. In excel raw data file, the average relative expression for MDR1 in the controls is > 1 whereas in the figure it is reported to be < 1. Could the authors comment this discrepancy and eventually do the appropriate correction in the figures ? Furthermore, in the figure 1E and 1F the relative expression of LAPTM4B in the controls should be the same, why do the authors reported a different values?

Validity of the findings

The conclusions of the study are supported by results but some additional experiments should give an additional validity to the findings.

·

Basic reporting

The manuscript by Rehman and colleagues highlights that in peripheral blood of breast, ovarian, prostate and colon cancer patients the expression levels of HIF-1α, MDR1 and LAPTM4B are higher than those of healthy counterparts. The authors also show the correlation between HIF-1α, MDR1 and LAPTM4B expression with advanced tumor stage, metastasis and chemotherapy experienced group in peripheral blood of the solid tumor patient’s samples.
The presented paper is clearly written but some main issues have to be clarified in order to validate the proposed thesis in a more accurate and in-depth way.

Experimental design

no comment

Validity of the findings

no comment

Additional comments

Major points:

1. In tables 1-5 are reported the correlations of HIF-1α, MDR1 and LAPTM4B expression with clinico-pathological features of patients samples and the analysis reveals differences in the expression levels of these genes. Can the authors classify the patients for HIF-1α, MDR1 and LAPTM4B levels and specific mutation in oncogenes (e.g. p53, kras, tgfb signalling, etc)?
2. The authors should mention the survival data of patients with solid tumors in correlation with HIF-1 α, MDR1 and LAPTM4B expression levels.
3. As further validation of the overexpression of these genes in tumor development, the authors should perform immunohistochemical analysis on tumor specimens.
4. It is known that not only the ubiquitary HIF-1α is involved in the mediation of the cellular response to hypoxia but also the HIF-2α is a key regulator of this process. Moreover, the HIF-2α gene is commonly analyzed by qPCR, while the variation in the expression of HIF-1α is usually analyzed by western blotting. Have the authors tried to analyze the correlation between HIF-2α and MDR1 and LAPTM4B expression levels?
5. The high expression levels of the HIF-1α, MDR1 and LAPTM4B where found by analysing the whole peripheral blood. Is there a specific expression of such genes in particular cells subpopulation? Are these mRNA presents in circulating microvescicles/exosomes?

---

## Round 0.2 · accepted · Accept

The reviewers are all favorable to accept your revised manuscript because all raised criticisms have been addressed.

# Reviewer 2 ·

Basic reporting

No comment

Experimental design

No comment

Validity of the findings

No comment

Additional comments

The authors well addressed the all raised questions

·

Basic reporting

The authors have satisfactorily responded to all my questions and made the necessary changes to the manuscript.

Experimental design

The authors have satisfactorily responded to all my questions and made the necessary changes to the manuscript.

Validity of the findings

The authors have satisfactorily responded to all my questions and made the necessary changes to the manuscript.

Additional comments

The authors have satisfactorily responded to all my questions and made the necessary changes to the manuscript.